# UniTok: A Unified Tokenizer for Visual Generation and Understanding

**Chuofan Ma**[1,2]    **Yi Jiang**[2†]    **Junfeng Wu**[2,3]    **Jihan Yang**[1]

**Xin Yu**[1]    **Zehuan Yuan**[2*]    **Bingyue Peng**[2]    **Xiaojuan Qi**[1†*]

[1]The University of Hong Kong    [2]ByteDance Inc.

[3]Huazhong University of Science and Technology

## Abstract

Visual generative and understanding models typically rely on distinct tokenizers to process images, presenting a key challenge for unifying them within a single framework. Recent studies attempt to address this by connecting the training of VQVAE (for autoregressive generation) and CLIP (for understanding) to build a unified tokenizer. However, directly combining these training objectives has been observed to cause severe loss conflicts. In this paper, we show that reconstruction and semantic supervision do not inherently conflict. Instead, the underlying bottleneck stems from limited representational capacity of discrete token space. Building on these insights, we introduce UniTok, a unified tokenizer featuring a novel multi-codebook quantization mechanism that effectively scales up the vocabulary size and bottleneck dimension. In terms of final performance, UniTok sets a new record of 0.38 rFID and 78.6% zero-shot accuracy on ImageNet. Besides, UniTok can be seamlessly integrated into MLLMs to unlock native visual generation capability, without compromising the understanding performance. Additionally, we show that UniTok favors *cfg-free generation*, reducing gFID from 14.6 to 2.5 on ImageNet 256×256 benchmark. GitHub: https://github.com/FoundationVision/UniTok.

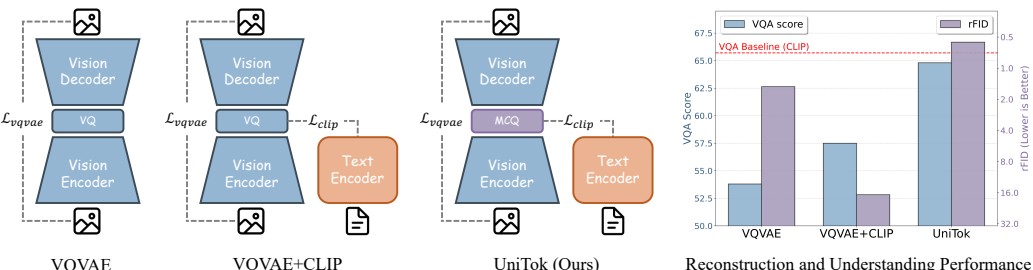

Figure 1: **The major challenge in unified tokenizer training.** CLIP supervision cannot be easily incorporated into VQVAE training – This provides only marginal improvements in understanding performance, while drastically degrading reconstruction FID.

## 1 Introduction

The advent of GPT-4o [33] highlights the immense potential of Multimodal Large Language Models (MLLMs) with native visual generation capabilities [12, 46, 62, 73, 58]. These unified models offer precise control in multimodal interactions, enabling exceptional fluency in tasks such as multi-turn image editing and visual in-context learning. However, a fundamental dilemma remains in the choice of visual tokenizers for unified MLLMs – e.g., the CLIP [38, 71] tokenizer excels in multimodal understanding but complicates generative modeling due to its high-dimensional, continuous feature

---

*: Corresponding authors;    †: Project lead.

39th Conference on Neural Information Processing Systems (NeurIPS 2025).

space; Conversely, the discrete VQVAE [8] tokenizer fits autoregressive generation but struggles to capture essential semantics for understanding [62].

In this work, we aim to design a unified visual tokenizer to bridge the gap in multimodal generation and understanding. Intuitively, this can be achieved by integrating CLIP supervision into VQVAE training, resulting in a discrete tokenizer capturing both fine-grained details and high-level semantics. However, we empirically find this training recipe confronts severe convergence issues [61] and largely falls behind the CLIP baseline in multimodal understanding (Figure 1). While prior studies commonly attribute these challenges to conflicts between semantic and pixel-level feature learning [61, 37, 58], recent progress in visual generation suggests the opposite, showing semantic regularization could benefit tokenizers in reconstruction-oriented training [63, 4, 19]. Such disparity motivates us question: **Do reconstruction and semantic losses truly conflict in tokenizer training?**

To study the problem, we conduct a comprehensive ablation on the unified tokenizer training paradigm (Figure 3), which yields several intriguing findings: First, we show that removing reconstruction supervision, which leads to a vector-quantized CLIP model, does not improve understanding performance compared to the unified tokenizer. This observation indicates that the performance gap between unified and CLIP tokenizers mainly arises from vector quantization, rather than conflicts between learning objectives; Further analysis reveals that this gap is driven by two key factors: token factorization, which projects tokens into a lower-dimensional space for code index lookup [65], and discretization. These operations are essential for vector quantization but inevitably compromise the expressiveness of visual tokens. We thus argue that **the primary bottleneck of unified tokenizers lies in the limited representational capacity of discrete token space**.

In light of the issue, we consider expanding the vocabulary size and latent code dimension, which allows for a closer approximation of the continuous feature space. However, extensive studies have shown that doing so could result in low codebook utilization [74, 65] and diminishing performance gains [67]. To address this, we introduce multi-codebook quantization to partition the visual token into several chunks, each discretized using a small, separate sub-codebook, akin to the multi-head attention mechanism [54]. This design exponentially scales the vocabulary size with the number of sub-codebooks, while avoiding the optimization problems of large monolithic codebooks. Besides, we replace traditional linear projection layers with adapted attention modules for token factorization, which is observed to consistently improve training stability and understanding performance.

Building upon these techniques, we train a unified tokenizer called UniTok to bridge visual generation and understanding. Through extensive experiments, we demonstrate that UniTok achieves comparable or even better performance to domain-specific tokenizers: On ImageNet evaluation, UniTok records an impressive 0.38 reconstruction FID and 78.6% zero-shot accuracy at 256×256 resolution; In building unified MLLMs, UniTok enables the MLLM with native visual generation capabilities while maintaining decent understanding performance. It outperforms the Liquid [59] baseline with a VQGAN tokenizer by 5.5% on VQAv2 [13], 9.2% on TextVQA [42], and 339 points on MME [64]; In addition, we demonstrate that semantic supervision leads to improved latent space structure for autoregressive generation, i.e., for class-conditional image generation on ImageNet 256×256, UniTok significantly reduces generation FID *without classifier-free guidance* from 14.6 to 2.5 under the LlamaGen [43] framework, which aligns with recent findings in diffusion modeling [19, 4, 63].

## 2   Related Work

**Image Tokenization for Generation.**   In the domain of visual generation, image tokenization plays an important role in encoding raw pixels into compact latent features for generative modeling [53, 39]. Among a variety of tokenizers, the vector-quantized tokenizer [53] is favored for its discrete latent space and compatibility with autoregressive or masked generative models [48, 43, 3, 66]. The pioneering work VQVAE [53] initially introduced the concept of discretizing continuous tokens by mapping them to the nearest neighbors in a learnable codebook. Built on this, VQGAN [8] added perceptual loss [72] and discriminator loss [16] to improve the reconstruction quality. ViT-VQGAN [65] subsequently advanced the framework with the transformer architecture. In recent literature, considerable efforts have been devoted to developing better quantization methods such as residual quantization [18] and lookup-free quantization [67], which also constitute a focal point of this paper.

**Image Tokenization for Understanding.**   The unprecedented success of large language models (LLMs) [57, 1, 51, 47] has catalyzed the development of multimodal large language models (MLLMs)

[28, 25, 31]. As a critical component of MLLMs, the selection of an effective vision tokenizer has been the subject of extensive study [55, 49]. A common choice of the vision tokenizer is the pretrained CLIP model [38], which undergoes alignment with language during its pretraining phase. While self-supervised learning models, such as DINOv2 [34], are shown to be advantageous at region-level tasks [30]. However, these tokenizers predominantly encode images into a continuous feature space, presenting challenges for uniformly modeling both vision and text tokens. To address this, some works have explored discretizing CLIP tokens [10] or employing VQVAE encoders [27, 62]. Yet, these methods have been observed to substantially impair understanding performance of MLLMs.

**Unified Vision-Language Models.** The rise of MLLMs is not limited to the realm of visual understanding. Recent advancements have witnessed an increasing focus on unifying visual generation and understanding within one MLLM [7, 60, 46, 73, 62, 50, 21]. Specifically, a line of works employs continuous visual tokenizers for image encoding, and leverages pretrained diffusion models for image synthesis [7, 11, 44]. This approach inevitably increases model complexity and disconnects the visual sampling process from the MLLM. In contrast, another stream of research adopts VQVAE models to encode images into discrete tokens [46, 56, 62, 61, 59]. These tokens are subsequently modeled using the same cross-entropy loss that is applied to text tokens, facilitating a unified approach to multimodal learning. However, as reconstruction-oriented VQVAE does not naturally align with the LLM token space, these models typically suffer from degraded visual comprehension capabilities. Our research aligns with the second approach, with a particular focus on the tokenizer design that is suitable for both generation and understanding tasks.

## 3 Method

In this section, we introduce UniTok, a unified tokenizer well-suited for both visual generation and understanding tasks. We start with a unified training recipe that integrates reconstruction (VQVAE) and semantic (CLIP) supervisions (Section 3.1). However, we find that simply combining both training objectives leads to severe performance degradation, which can be mainly attributed to limited representational capacity of discrete tokens (Section 3.2). To this end, we propose multi-codebook quantization and attention projection to enhance the latent feature space and derive unified visual representations (Section 3.3). An overview of the framework is presented in Figure 2.

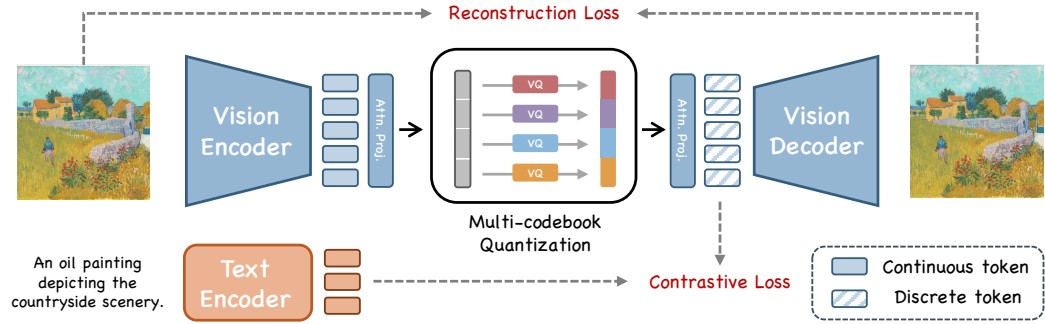

Figure 2: **An overview of UniTok.** The tokenizer is trained to reconstruct the input image while aligning its discrete latent features with the text caption. For vector quantization, each visual token is split into multiple chunks, which then undergo code index lookup on corresponding sub-codebooks.

### 3.1 Unified Supervision

Visual generative and understanding models typically impose distinct demands on the visual tokenizers. For instance, generation emphasizes precise encoding of the visual signals, whereas understanding prioritizes capturing high-level semantics. To accommodate both requirements, we jointly train the tokenizer with *(i)* a VQVAE-based reconstruction loss to preserve low-level information, and *(ii)* an image-text contrastive loss that enhances high-level semantics of the features.

To be specific, the VQVAE-based loss term $\mathcal{L}_{\mathrm{recon}}$ consists of a pixel-level reconstruction loss $\mathcal{L}_{\mathrm{R}}$, a perceptual loss $\mathcal{L}_{\mathrm{P}}$ based on the LPIPS metric [72], a discriminator loss $\mathcal{L}_{\mathrm{G}}$ to enhance reconstruction

fidelity [16], and a vector quantization loss $\mathcal{L}_{\text{VQ}}$ to minimize distance between the encoder output and its nearest code entry. It is denoted as:

$$\mathcal{L}_{\text{recon}} = \mathcal{L}_{\text{R}} + \lambda_{\text{VQ}}\mathcal{L}_{\text{VQ}} + \lambda_{\text{P}}\mathcal{L}_{\text{P}} + \lambda_{\text{G}}\mathcal{L}_{\text{G}}, \tag{1}$$

where $\lambda$ is the weight factor for the corresponding loss term. The image-text contrastive loss term $\mathcal{L}_{\text{contra}}$ is basically the same as in CLIP [38]. Therefore, the final loss term can be written as:

$$\mathcal{L} = \mathcal{L}_{\text{recon}} + \lambda_{\text{contra}}\mathcal{L}_{\text{contra}}. \tag{2}$$

We simply choose $\lambda_{\text{contra}} = 1$ in this paper.

## 3.2 Quantization Bottleneck

Despite being augmented with CLIP supervision, we find that the unified tokenizer exhibits unsatisfactory performance in visual understanding tasks, significantly lagging behind the commonly used CLIP tokenizer. To figure out the underlying cause of this underperformance, we break down the key components involved in training a unified tokenizer, as illustrated in Figure 3. Starting with the CLIP baseline, we provide a step-by-step walk-through of all changes in following paragraphs.

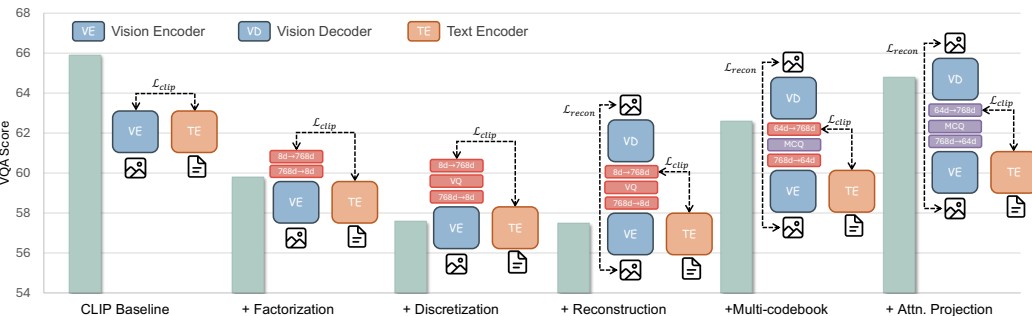

Figure 3: **Roadmap from CLIP to UniTok.** It is observed that major degradation in understanding performance comes from token factorization and discretization, rather than reconstruction supervision. The proposed multi-codebook quantization and attention projection effectively address this by scaling up the vocabulary size and bottleneck dimension. The VQA score is measured using the average score across the VQAv2, GQA, TextVQA, and POPE benchmarks. All tokenizers are trained from scratch on 512m image-text pairs from DataComp.

**Factorization.** Modern VQ-tokenizers typically project continuous tokens to a lower-dimensional latent space for code index lookup (e.g. from 768-d to 8-d), known as token factorization [65]. This increases the relative density of codes by compressing the latent code space, thereby reducing quantization error. To evaluate the impact of factorization in CLIP training, we add two linear projection layers on top of the CLIP vision encoder (right before average pooling), which transforms tokens from 768-d to 16-d and then back to 768-d. Notably, vector quantization and reconstruction supervision are not included at this stage. Surprisingly, it turns out that this channel compression operation significantly compromises the expressiveness of tokens, leading to severe performance degradation in downstream VQA tasks.

**Discretization.** Based on the implementation described above, we further introduce vector quantization to CLIP training, which maps factorized tokens to their nearest code entries. Compared to language tokenizers with vocabularies exceeding 200k entries, the vocabulary size of modern VQ-tokenizers is markedly smaller (i.e., typically ranging from 4k to 16k). Mapping continuous tokens to such a small codebook results in considerable information loss. This is validated in our experiment, which demonstrates that discretizing the factorized tokens with a 16k codebook causes an average accuracy drop of 2.1 in VQA tasks.

**Reconstruction Supervision.** Finally, we integrate reconstruction losses into the training process to build a unified tokenizer, as outlined in Section 3.1. Previous literature suggests that loss conflict between VQVAE and CLIP is a major cause of performance degradation in joint training [61]. We observe a similar phenomenon where joint training results in sub-optimal ImageNet zero-shot classification accuracy and reconstruction FID compared to specialized training. However, surprisingly, we find that this degradation has negligible impacts on downstream understanding performance.

Moreover, the degradation in classification accuracy and reconstruction FID diminishes after we improve the quantization methods (detailed in the next section). Based on these observations, we speculate that the perceived loss conflict is only a superficial issue, and the primary cause of the underperformance lies in the limited representational capacity of discrete tokens.

## 3.3 UniTok

A straightforward solution to breaking the quantization bottleneck could be increasing the codebook size and the latent code dimension. However, current studies on VQVAE tokenizers suggest that there is diminishing gain in scaling and the performance saturates after the codebook size reaches 16k [67, 43]. Continuing expansion results in a substantial portion of codes being rarely used or becoming 'dead' during training, which negatively impacts downstream task performance [65]. To address this, we propose multi-codebook quantization and attention projection in the following paragraphs.

**Multi-codebook quantization (MCQ)** discretizes the latent tokens with a set of independent codebooks. Specifically, the latent vector $f \in \mathbb{R}^d$ is first evenly split into $n$ chunks $\{f_1, f_2, ..., f_n\}$, where $f_i \in \mathbb{R}^{\frac{d}{n}}$. The subsequent quantization process is denoted as:

$$\hat{f} = \text{Concat}\left(\mathcal{Q}\left(Z_1, f_1\right), \mathcal{Q}\left(Z_2, f_2\right), ..., \mathcal{Q}\left(Z_n, f_n\right)\right) \tag{3}$$

where $\hat{f}$ is the discretized latent vector, $\mathcal{Q}$ is the code index lookup operation, and $Z_i$ is $i$-th sub-codebook. Compared to conventional quantization methods, the proposed MCQ effectively scales up the vocabulary size. For instance, by increasing the number of sub-codebooks from 1 to 4, and suppose each sub-codebook contains 16k code entries, the theoretical vocabulary size exponentially increases from $2^{14}$ to $2^{56}$ (i.e., there are up to $2^{14 \times 4}$ possible combinations of codes for each token). As the size of each individual codebook remains constant, it circumvents the optimization problem associated with large codebooks. Besides, the dimensionality of the latent codes also scales proportionally with the number of codebooks (i.e., increasing from 16-d to 64-d in this case), which further enhances the representational capacity of discrete representations.

**Discussions.** MCQ shares a similar concept with residual quantization (RQ) [18] in using multiple codes to quantize a token, but differs fundamentally in design philosophy: RQ follows a coarse-to-fine quantization order, whereas MCQ adopts a divide-and-conquer strategy. This distinction gives MCQ unique advantages when operating in high-dimensional latent spaces, where codes tend to become increasingly sparse. For instance, with a latent dimension of 64-d, we observe that MCQ's quantization loss is 15 to 45 times lower than that of RQ. This is because MCQ partitions the original latent space into multiple low-dimensional subspaces for quantization. Our ablation study in Table 7 further confirms the superiority of MCQ in unified tokenizer training.

**Attention projection.** Existing VQ methods usually employ linear or convolutional projection layers for token factorization. But as shown in Figure 3, this over-simplified design fails to preserve rich semantics when compressing the feature dimensions, leading to degraded understanding performance. To alleviate this problem, we suggest adapting the multi-head attention modules for factorization. Specifically, instead of concatenating features from multiple heads after the attention calculation, we replace the concatenation operation with average pooling to realize channel compression. Figure 6 provides a detailed illustration of the adaptation. Despite its simplicity, we find this design effectively strengthens the representational power of factorized tokens and stabilizes training.

## 3.4 Unified MLLM

We proceed to develop a unified multimodal model with UniTok. Particularly, we leverage the unified framework introduced in Liquid [59], which models (discrete-valued) vision and language sequences with a universal next-token prediction loss. But instead of learning the visual codebook from scratch, we reuse code embeddings of UniTok by projecting them to the MLLM token space with an MLP projector. Notably, despite UniTok encodes an image into $H \times W \times K$ codes (where $K$ represents the number of sub-codebooks), we simplify this for MLLM input by merging every $K$ consecutive codes into a single visual token. Similarly, when it comes to visual token prediction, we make each token autoregressively predict the next $K$ codes, using a depth transformer head as implemented in RQ-Transformer [18] and VILA-U [61]. This design maintains efficiency for visual generation in the context of multi-codebooks.

# 4 Experiments

## 4.1 Implementation Details

**Tokenizer Setup.** Leading VQVAE tokenizers predominantly adopt the CNN architecture, while ViT is preferred in CLIP training for its scalability. To take advantage of both, we choose a hybrid architecture, ViTamin-L/16 [5], to instantiate UniTok. We configure UniTok with eight sub-codebooks, each containing 4,096 code entries and a latent dimension set to 8-d (the global latent dimension is thus 64-d). The discriminator is initialized with pretrained DINOv2-S [34]. We train the tokenizer for one epoch on the public dataset DataComp-1B [9] consisting of 1.28B image-text pairs, with all images resized to $256 \times 256$ resolution and a global batch size of 16k. The learning rate is set to 1e-3 for the tokenizer and 2e-4 for the discriminator. Besides, we prepare two settings for evaluation: one with pretrained CLIP weight initialization and one with random initialization (the default setting).

**MLLM Setup.** We instantiate a unified MLLM described in Section 3.4 with the Llama-2-7B base model [52]. Following Liquid, we first pretrain the model on a mix of multimodal data, which is composed of 10M language data from DCLM [22], 30M internal MidJourney-style synthetic data, and 30M re-captioned image-text pairs from COYO [32] and Laion [41]. Subsequently, we finetune the model on 1.5M text-to-image data and 1.5M multimodal instruction tuning data introduced in Mini-Gemini [23]. Specifically, the learning rate is set to 5e-5 in the pretraining stage and 2e-5 in the finetuning stage. For visual understanding evaluation, we report results on standard VQA benchmarks including VQAv2 [13], GQA [14], TextVQA [42], POPE [24], MME [64], and MM-Vet [70]. For visual generation evaluation, we report results on GenAI-Bench [26] and MJHQ-30K [20].

## 4.2 Tokenizer Comparison

We benchmark UniTok on ImageNet using two primary metrics: Fréchet Inception Distance (FID) to evaluate reconstruction quality, and top-1 zero-shot accuracy to assess image-text alignment. The results are presented in Table 1. To provide a fair comparison with tokenizers trained on small datasets, we also train a version of UniTok on OpenImages [17] solely with reconstruction supervision. It can be seen that UniTok excels in reconstruction quality compared to both unified and domain-specific tokenizers, recording an impressive 0.38 rFID on ImageNet with $16\times$ downsampling ratio. As a discrete tokenizer, UniTok even surpasses the continuous VAE tokenizer from Stable Diffusion v2.1 [40], showcasing the superiority of the proposed multi-codebook quantization. For the perception performance, we observe that randomly initialized UniTok demonstrates suboptimal zero-shot classification accuracy. This is expected as current training schedule (i.e., one epoch on 1.28B samples) is insufficient for CLIP training to fully converge. It can be seen that initializing the model with pretrained CLIP weights largely alleviates the problem, boosting the zero-shot accuracy from 70.8% to 78.6%. In complement to quantitative results, we provide examples of reconstructed images in Figure 4.

Table 1: Comparison on ImageNet reconstruction FID and zero-shot classification accuracy. rFID is measured at 256×256 resolution with 16× downsample ratio. † indicates model using pretrained CLIP weights for initialization. ∗ indicates model trained on OpenImages.

| Method | #Tokens | rFID ↓ | Accuracy |
|---|---|---|---|
| *VQVAE Model* | | | |
| VQ-GAN* [8] | 256 | 4.98 | – |
| RQ-VAE [18] | 256 | 1.30 | – |
| VAR* [48] | 680 | 0.90 | – |
| UniTok* | 256 | 0.33 | – |
| *CLIP Model* | | | |
| CLIP [38] | 256 | – | 76.2 |
| SigLIP [71] | 256 | – | 80.5 |
| ViTamin [5] | 256 | – | 81.2 |
| *Unified Model* | | | |
| TokenFlow† [37] | 680 | 1.37 | – |
| VILA-U† [61] | 256 | 1.80 | 73.3 |
| UniTok | 256 | 0.41 | 70.8 |
| UniTok† | 256 | 0.38 | 78.6 |

## 4.3 Class-Conditional Image Generation

Recent studies on diffusion models indicate that injecting semantics into VAE training leads to a better-structured latent space, considerably enhancing guidance-free generation performance [63, 4, 19]. To evaluate whether UniTok possesses similar properties, we test it within the LlamaGen framework for class-conditional image generation. As shown in Table 2, UniTok reduces the FID by 12.11 compared to the VQGAN baseline in CFG-free generation, under the same generator setup. This implies that UniTok learns a more structured code distribution, benefiting autoregressive modeling.

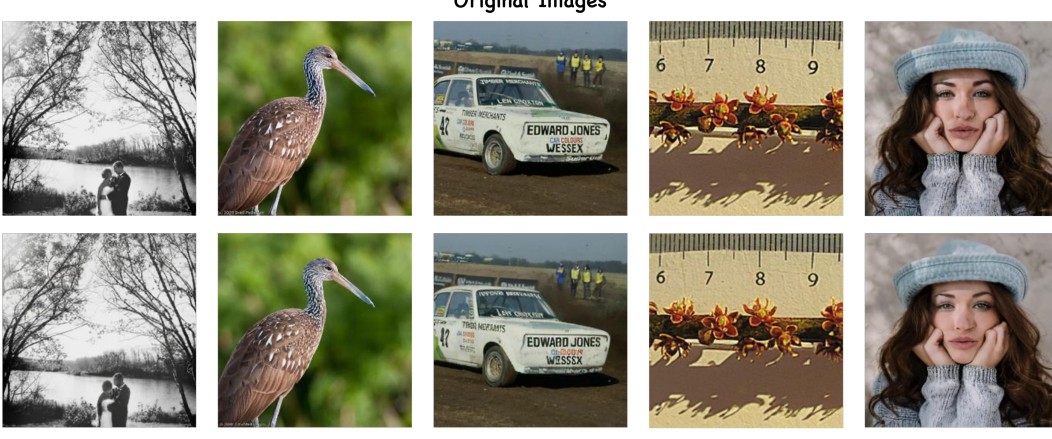
Original Images

Reconstructed Images

Figure 4: Qualitative results on image reconstruction in a resolution of $256 \times 256$.

Table 2: Class-conditional image generation results on ImageNet 256×256. [†]: VQGAN tokenizer from LlamaGen. [‡]: Images are generated at 384×384 resolution and then resized to 256×256 for evaluation. 'Pre.': precision; 'Rec.': recall; 'CFG': classifier-free-guidance.

| Tokenizer | rFID | Generator | #Params. | Generation w/o CFG | | | | Generation w/ CFG | | | |
|---|---|---|---|---|---|---|---|---|---|---|---|
| | | | | gFID↓ | IS↑ | Pre. | Rec. | gFID↓ | IS↑ | Pre. | Rec. |
| *Diffusion Models* | | | | | | | | | | | |
| SD-VAE [40] | 0.61 | DiT [35] | 675M | 9.62 | 121.5 | 0.67 | 0.67 | 2.27 | 278.2 | 0.83 | 0.57 |
| VAVAE [63] | 0.28 | LightningDiT [63] | 675M | 2.17 | 205.6 | 0.77 | 0.65 | 1.35 | 295.3 | 0.79 | 0.65 |
| *Masked Generative Models* | | | | | | | | | | | |
| LFQ [67] | 0.9 | MAGVIT-v2 [67] | 307M | 3.07 | 213.1 | – | – | 1.91 | 324.3 | – | – |
| TiTok-L [69] | 2.21 | MaskGIT [3] | 177M | 3.15 | 173.0 | – | – | 2.77 | 199.8 | – | – |
| *Autoregressive Models* | | | | | | | | | | | |
| VQGAN[†] | 2.19 | LlamaGen[‡] [43] | 1.4B | 14.65 | 86.3 | 0.63 | 0.68 | 2.34 | 253.9 | 0.81 | 0.60 |
| UniTok (Ours) | 0.41 | LlamaGen [43] | 1.4B | **2.51** | **216.7** | 0.82 | 0.57 | 2.77 | 227.5 | 0.81 | 0.57 |

## 4.4 Unified Understanding and Generation

**Understanding Performance.** We evaluate the understanding performance of UniTok on diverse VQA benchmarks in Table 3. Our unified MLLM showcases clear advantages when compared to other unified models that also utilize a discrete visual tokenizer. Specifically, UniTok significantly outperforms the Chameleon model, which relies on a traditional VQVAE tokenizer, by 7.2% higher accuracy on VQAv2. Additionally, it surpasses VILA-U, another model with a unified tokenizer, by 3.3% in accuracy on the TextVQA benchmark and by a notable margin of 112 points on the MME-Perception scores. Furthermore, we can see that UniTok largely narrows the performance gap with MLLMs that incorporate continuous visual tokenizers. These strong results confirm the candidacy of UniTok as a unified visual tokenizer for multimodal models.

**Generation Performance.** Table 4 presents the text-to-image generation performance of our unified MLLM on the GenEval benchmark. We show that UniTok not only outperforms most of the unified MLLMs, but also demonstrates competitive performance against domain experts (diffusion models) trained on billions of images. Besides, UniTok achieves non-trivial improvements over Liquid while using exactly the same set of text-to-image training data, highlighting the importance of a unified tokenizer. We also provide results on GenAI-Bench in Table 10 and Table 11 in Appendix.

We further evaluate the quality of images generated by our model on the MJHQ-30K benchmark, details of which are presented in Table 5. Notably, as this benchmark primarily relies on the FID score

Table 3: Comparison with unified multi-modal large language models on VQA benchmarks.

| Method | LLM | Token Type | Res. | VQAv2 | GQA | TextVQA | POPE | MME | MM-Vet |
|---|---|---|---|---|---|---|---|---|---|
| Emu [45] | Llama-13B | Continuous | 224 | 52.0 | - | - | - | - | - |
| LaVIT [15] | Llama-7B | Continuous | 224 | 66.0 | 46.8 | - | - | - | - |
| DreamLLM [7] | Vicuna-7B | Continuous | 224 | 72.9 | - | 41.8 | - | - | 26.6 |
| Unified-IO 2 [29] | 6.8B from scratch | Continuous | 384 | 79.4 | - | - | 87.7 | - | - |
| Janus [58] | DeepSeek-1.3B | Continuous | 384 | 77.3 | 59.1 | - | 87.0 | 1338 | 34.3 |
| CM3Leon [68] | 7B from scratch | Discrete | 256 | 47.6 | - | - | - | - | - |
| LWM [27] | Llama-2-7B | Discrete | 256 | 55.8 | 44.8 | 18.8 | 75.2 | - | - |
| Show-o [62] | Phi-1.5-1.3B | Discrete | 256 | 59.3 | 48.7 | - | 73.8 | 948 | - |
| Chameleon [46] | 34B from scratch | Discrete | 512 | 69.6 | - | - | - | - | - |
| Liquid [59] | Gemma-7B | Discrete | 512 | 71.3 | 58.4 | 42.4 | 81.1 | 1119 | - |
| VILA-U [61] | Llama-2-7B | Discrete | 256 | 75.3 | 58.3 | 48.3 | 83.9 | 1336 | 27.7 |
| UniTok | Llama-2-7B | Discrete | 256 | 76.8 | 61.1 | 51.6 | 83.2 | 1448 | 33.9 |

Table 4: Comparison with other visual generation methods on the GenEval benchmark.

| Method | Type | #Data | Single Obj. | Two Obj. | Counting | Colors | Position | Color Attri. | Overall↑ |
|---|---|---|---|---|---|---|---|---|---|
| SD v2.1 [39] | Diffusion | 2000M | 0.98 | 0.51 | 0.44 | 0.85 | 0.07 | 0.17 | 0.50 |
| SD-XL [36] | Diffusion | 2000M | 0.98 | 0.74 | 0.39 | 0.85 | 0.15 | 0.23 | 0.55 |
| DALL-E 3 [2] | Diffusion | – | 0.96 | 0.87 | 0.47 | 0.83 | 0.43 | 0.45 | 0.67 |
| Show-o [62] | Discrete Diff. | 36M | 0.95 | 0.52 | 0.49 | 0.82 | 0.11 | 0.28 | 0.53 |
| | | 2.0B | 0.98 | 0.80 | 0.66 | 0.84 | 0.31 | 0.50 | 0.68 |
| LWM [27] | Autoregressive | – | 0.93 | 0.41 | 0.46 | 0.79 | 0.09 | 0.15 | 0.47 |
| Janus [58] | Autoregressive | – | 0.97 | 0.68 | 0.30 | 0.84 | 0.46 | 0.42 | 0.61 |
| Liquid [59] | Autoregressive | 30M | 0.98 | 0.73 | 0.32 | 0.76 | 0.17 | 0.37 | 0.55 |
| UniTok | Autoregressive | 30M | 0.99 | 0.71 | 0.36 | 0.79 | 0.26 | 0.45 | 0.59 |

for evaluation, high-resolution images are preferred because they potentially capture more fine-grained details. Despite this makes FID across different resolutions less comparable, we show that our model achieves impressive performance even at the the smallest resolution, showcasing its ability to generate high-quality, detail-rich images.

We present some examples of the images generated by our model in Figure 5, using text prompts sampled from MJHQ-30K. The visualization results demonstrate our model is capable of synthesizing photorealistic and visually appealing images. Moreover, the model is able to comprehend a wide spectrum

Table 5: Results on MJHQ-30K.

| Method | Type | Res. | FID↓ |
|---|---|---|---|
| SD-XL [36] | Diffusion | 1024 | 9.55 |
| PixArt [6] | Diffusion | 1024 | 6.14 |
| Playground [20] | Diffusion | 1024 | 4.48 |
| Liquid [59] | Autoregressive | 512 | 5.47 |
| Janus [58] | Autoregressive | 384 | 10.10 |
| LWM [27] | Autoregressive | 256 | 17.77 |
| Show-o [62] | Discrete Diff. | 256 | 15.18 |
| VILA-U [61] | Autoregressive | 256 | 12.81 |
| UniTok | Autoregressive | 256 | 7.46 |

of concepts, such as 'Vincent van Gogh painting style' and 'bitcoin', and flexibly combine these concepts to synthesize creative images.

### 4.5 Ablation Studies

**Impact of Supervision Types.** To ablate the impact of contrastive and reconstruction losses in UniTok training, we conduct experiments on tokenizers trained with different supervision types, as shown in Table 6. It is worth noting that all the tokenizers are vector-quantized even though some do not have reconstruction supervision. First, we show that reconstruction-oriented tokenizer significantly lags behind tokenizers with contrastive supervision in visual understanding performance. This observation evidences the limitations of traditional VQVAE. Second, we demonstrate that reconstruction and contrastive training objectives do not inherently conflict, or can be addressed by enhancing discrete feature space. With multi-codebook quantization, the jointly trained tokenizer not only exhibits understanding performance on par with the tokenizer trained solely with contrastive loss, but also slightly improves generation performance over the reconstruction-oriented tokenizer.

**MCQ v.s. RQ.** Following discussions in Section 3.3, we provide an apple-to-apple comparison between multi-codebook quantization and residual quantization in Table 7. For fair comparisons, we

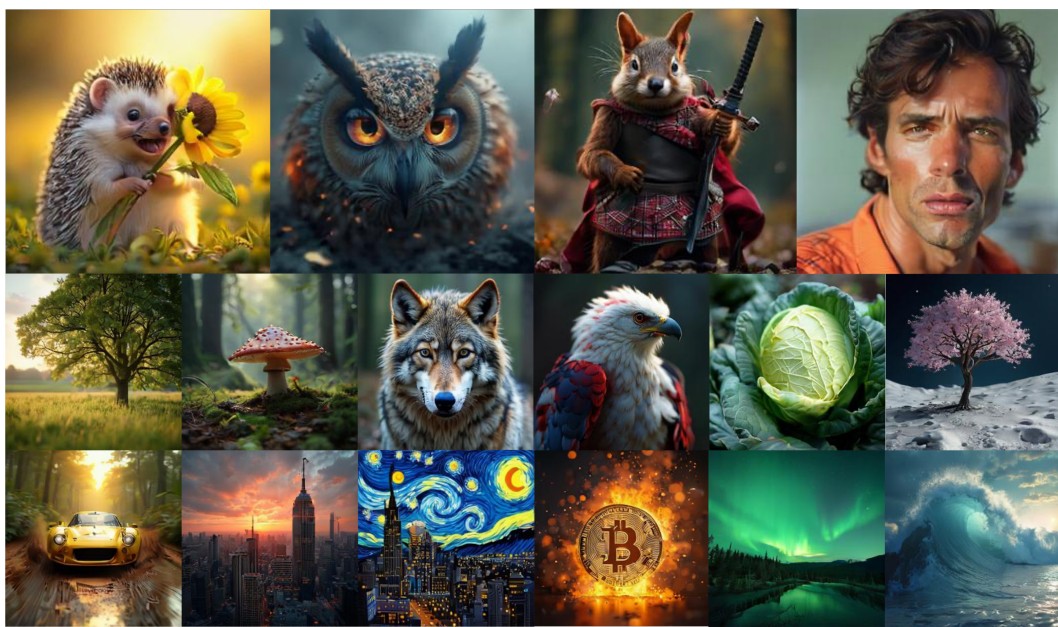

Figure 5: Images generated in a resolution of $256 \times 256$ with our unified MLLM.

Table 6: Impact of different supervision types on downstream generation and understanding performance. The rFID and gFID are measured on the ImageNet ($256 \times 256$) validation set. LlamaGen-L [43] is adopted as the generator for gFID evaluation.

| Supervision | Generation | | Understanding | | | | | |
|---|---|---|---|---|---|---|---|---|
| | rFID ↓ | gFID ↓ | VQAv2 | GQA | SciQA | TextVQA | POPE | MME |
| Contrastive | – | – | 68.95 | 56.89 | 65.64 | 49.89 | 82.34 | 1373 |
| Reconstruction | 0.82 | 3.59 | 56.33 | 47.53 | 63.26 | 43.65 | 77.09 | 902 |
| Recon. + Contra. | 0.72 | 3.26 | 69.14 | 56.06 | 65.25 | 49.22 | 81.42 | 1333 |

directly implement RQ on our codebase, keeping all the training settings the same as UniTok. Both tokenizers are trained on a 512M subset of DataComp-1B. Notably, unlike MCQ, RQ by default uses a shared large codebook. To keep the global codebook size the same as UniTok, we set the codebook size of RQ to 32768. It can be seen that RQ demonstrates inferior reconstruction performance and lower classification accuracy compared to MCQ under the high bottleneck dimension (64-d) setting.

**Number of Sub-Codebooks.** To gain deeper insights into multi-codebook quantization, we evaluate how tokenizer performance changes with the number of sub-codebooks in Table 8. Specifically, the size of a codebook is denoted as $A \times B$, where $A$ is the number of sub-codebook and $B$ is the size of sub-codebook. For rFID evaluation, we train the tokenizer solely with reconstruction loss on OpenImages [17], and evaluated it on ImageNet ($256 \times 256$) validation set. While for ImageNet zero-shot accuracy evaluation, the tokenizer is trained on DataComp-1B 128m subset using only contrastive loss. Given a constant global codebook size, we see that increasing the number of sub-codebooks consistently improves reconstruction FID and classification accuracy. This indicates that MCQ generally benefits vector-quantized models, independent of the training objectives.

Table 7: MCQ v.s. RQ.

| Method | Code Shape | Code Dim. | rFID↓ | Accuracy |
|---|---|---|---|---|
| RQ | $16\times16\times8$ | 64 | 3.46 | 58.8 |
| MCQ | $16\times16\times8$ | 64 | 0.55 | 63.7 |

Table 8: Ablation on number of sub-codebooks.

| Codebook / Vocabulary | $1\times16384 / 2^{14}$ | $2\times8192 / 2^{26}$ | $4\times4096 / 2^{48}$ | $8\times2048 / 2^{88}$ |
|---|---|---|---|---|
| rFID ↓ | 1.50 | 0.98 | 0.54 | 0.33 |
| Accuracy | 41.0% | 43.9% | 44.7% | 46.1% |

**CLIP Weight Initialization.** We notice that higher ImageNet accuracy does not guarantee superior downstream performance. In Table 9, we ablate the impact of CLIP weight initialization on visual understanding performance. Specifically, we adopt the classic LLaVA framework for evaluation,

replacing the original CLIP tokenizer with UniTok while keeping all other the training settings unchanged. One tokenizer is initialized with the pretrained ViTamin-L-256 [5] weights, while the other is randomly initialized. To our surprise, UniTok that is trained from scratch surpasses the one initialized with pretrained CLIP weights, despite the latter actually achieves better zero-shot classification accuracy. This suggests downstream VQA performance may not be highly correlated with ImageNet classification accuracy. More importantly, it also implies that CLIP weight initialization may serve as a negative prior for unified tokenizers, as the unified visual feature space could drastically differ from CLIP feature space.

Table 9: Comparison of different initialization methods under the LLaVA framework. † indicates the model uses CLIP weights for initialization. We highlight the default setting of UniTok in gray.

| Tokenizer | VQAv2 | GQA | TextVQA | POPE | MME |
|---|---|---|---|---|---|
| UniTok† | 69.9 | 56.2 | 49.3 | 81.2 | 1331 |
| UniTok | 72.4 | 58.2 | 51.6 | 82.4 | 1392 |

## 5  Limitations and Conclusion

This paper studies unified visual tokenization for generation and understanding, which serves as the cornerstone of unified multimodal large language models. We investigate the training paradigm of unified tokenizers and identify that the current challenge in unification mainly arises from the limited representational power of discrete tokens. To address this limitation, we introduce multi-codebook quantization and attention projection to build a unified tokenizer called UniTok. We show that UniTok excels in downstream visual generation and understanding tasks. The ablation study further reveals that discriminative and generative representation learning does not inherently conflict. We hope our findings could inspire future research in this domain.

However, due to limited computational resources, UniTok is only trained for one epoch, which is not sufficient for CLIP-based semantic representation learning. We believe extending the training schedule could further benefit the tokenizer, especially in understanding performance.

**Acknowledgments** This work has been supported by the National Key R&D Program of China (Grant No. 2022YFB3608300), Hong Kong Research Grant Council - Early Career Scheme (Grant No. 27209621), General Research Fund Scheme (Grant No. 17202422, 17212923, 17215025) Themebased Research (Grant No. T45-701/22-R) and Shenzhen Science and Technology Innovation Commission (SGDX20220530111405040). We sincerely thank Seng Ye for the insightful discussions and valuable contributions to this project.

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

## A  Attention Projection Modules

Figure 6 illustrates the adaptations we made to the tradition MHA module, which enables channel compression and expansion in token factorization.

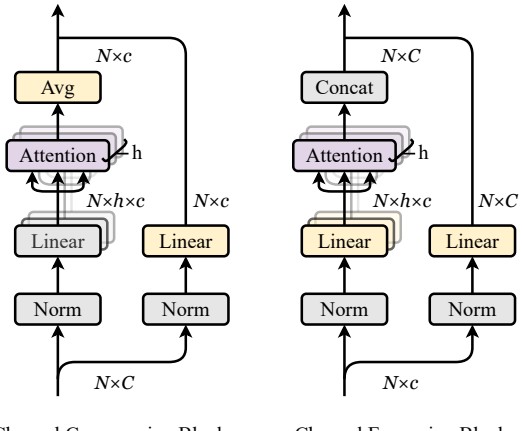

Channel Compression Block        Channel Expansion Block

Figure 6: Modified attention blocks for factorization. Modules in yellow indicate a change in the number of channels. $C$ and $c$ stand for the channel dimension, $h$ is the number of heads in the multi-head attention module. $C = h \times c$.

## B  More Generation Results

We provide the results on GenAI-Bench in Table 10 and Table 11. UniTok consistently delivers superior generation performance on this benchmark.

Table 10: Comparison with other visual generation methods on GenAI-Bench (basic prompts).

| Method | Type | #Training Images | Attribute↑ | Scene↑ | Relation↑ | | | Overall↑ |
| --- | --- | --- | --- | --- | --- | --- | --- | --- |
| | | | | | Spatial | Action | Part | |
| SD v2.1 [39] | Diffusion | 2000M | 0.80 | 0.79 | 0.76 | 0.77 | 0.80 | 0.78 |
| SD-XL [36] | Diffusion | 2000M | 0.84 | 0.84 | 0.82 | 0.83 | 0.89 | 0.83 |
| Midjourney v6 | Diffusion | – | 0.88 | 0.87 | 0.87 | 0.87 | 0.91 | 0.87 |
| DALL-E 3 [2] | Diffusion | – | 0.91 | 0.90 | 0.92 | 0.89 | 0.91 | 0.90 |
| Show-o [62] | Discrete Diff. | 36M | 0.72 | 0.72 | 0.70 | 0.70 | 0.75 | 0.70 |
| LWM [27] | Autoregressive | – | 0.63 | 0.62 | 0.65 | 0.63 | 0.70 | 0.63 |
| VILA-U [61] | Autoregressive | 15M | 0.78 | 0.78 | 0.77 | 0.78 | 0.79 | 0.76 |
| Liquid [59] | Autoregressive | 30M | 0.84 | 0.86 | 0.81 | 0.83 | 0.91 | 0.83 |
| UniTok | Autoregressive | 30M | 0.85 | 0.87 | 0.86 | 0.86 | 0.89 | 0.85 |

Table 11: Comparison with other visual generation methods on GenAI-Bench (advanced prompts).

| Method | Type | #Training Images | Count↑ | Differ↑ | Compare↑ | Logical↑ | | Overall↑ |
| --- | --- | --- | --- | --- | --- | --- | --- | --- |
| | | | | | | Negate | Universal | |
| SD v2.1 [39] | Diffusion | 2000M | 0.68 | 0.70 | 0.68 | 0.54 | 0.64 | 0.62 |
| SD-XL [36] | Diffusion | 2000M | 0.71 | 0.73 | 0.69 | 0.50 | 0.66 | 0.63 |
| Midjourney v6 | Diffusion | – | 0.78 | 0.78 | 0.79 | 0.50 | 0.76 | 0.69 |
| DALL-E 3 [2] | Diffusion | – | 0.82 | 0.78 | 0.82 | 0.48 | 0.80 | 0.70 |
| Show-o [62] | Discrete Diff. | 36M | 0.70 | 0.62 | 0.71 | 0.51 | 0.65 | 0.60 |
| LWM [27] | Autoregressive | – | 0.59 | 0.58 | 0.54 | 0.49 | 0.52 | 0.53 |
| VILA-U [61] | Autoregressive | 15M | 0.70 | 0.71 | 0.74 | 0.53 | 0.66 | 0.64 |
| Liquid [59] | Autoregressive | 30M | 0.76 | 0.73 | 0.74 | 0.46 | 0.74 | 0.65 |
| UniTok | Autoregressive | 30M | 0.76 | 0.76 | 0.79 | 0.46 | 0.73 | 0.67 |

## C   Impact Statement

This paper presents work whose goal is to advance the field of Machine Learning. There are many potential societal consequences of our work, none which we feel must be specifically highlighted here.

