# OpenReview forum: "UniTok: a Unified Tokenizer for Visual Generation and Understanding"
_NeurIPS.cc/2025/Conference — NeurIPS 2025 spotlight_

### Official Review · Reviewer_emrz · 2025-06-16

**Clarity:** 2
**Significance:** 3
**Originality:** 2
**Rating:** 4
**Confidence:** 4

**Summary:**

This work studies how to design a unified discrete tokeniser for both image understanding and generation tasks. It identifies the bottle necks of directly combining CLIP and VQVAE: improper compression from the network architecture, a single codebook. They made some improvements to enhance the representative capacity of the discrete token space. Experimental results verify the validity of the proposed method.

**Questions:**

Please see the weakness.

**Ethical Concerns:**

["NO or VERY MINOR ethics concerns only"]

**Final Justification:**

I maintain my initial score, as some of my concerns have been addressed.

**Limitations:**

Yes.

**Paper Formatting Concerns:**

None.

**Quality:**

3

**Strengths And Weaknesses:**

Strength
1. This paper studies an interesting topic and reveals that under proper design, the discrete tokenizer can achieve good performance on both understanding and generation tasks.
2. The paper is well oragnzied and  easy to follow.

Weakness
1. It seems that the comparison in Table 7 is unfair. RQ also support multi codes/groups, therefore, 1×32768  (K=1)is not fair for using the RQ baseline. Can you show the results for RQ(K=4 or 8?)?
2. The baseline in Table 2 is a bit old and weak.  Vavae [1] achieves much better FID (2.17 wo CFG /1.35 w CFG, only 675M parameters) than the cited baselines. If compared with this baseline, the performance of this work is not that good.
3. It's a bit strange to do the main understanding experiment using Liquid, which is not published yet by the DDL of NeurIPS.  VILA-U is a better choice (it provides the data and code).  Moreover, this setting can provide a controlled experiment for training data, which is well recognized critical for final performance.
4. Does the performance in Table 3 come from better training data?
5. Minor: It's better to include the performance on the GenEval [2] benchmark, because many unified models also cover it.

[1] Reconstruction vs. generation: Taming optimization dilemma in latent diffusion models, CVPR25
[2] GenEval: An Object-Focused Framework for Evaluating Text-to-Image Alignment, NeurIPS

---

> ### Author Rebuttal · Authors · 2025-07-31
>
> Dear Reviewer emrz,
>
> Thank you for your constructive feedback and thoughtful evaluation of our work. We hope our point-to-point response will address your concerns.
>
> **Q1: Fair comparison with RQ.**
>
> We apologize for any misconceptions caused. In Table 7, we do implement RQ with the multi-code setting (i.e., using 8 codes to represent a single token), consistent with the MCQ approach. For RQ, the codes are generated by recursively quantizing each token, so the number of codes is determined by the quantization depth (D) rather than the number of codebooks (K). Following the official RQ implementation, we set D = 8 and K = 1 for the experiments in Table 7. This results in 16 × 16 × 8 codes per image, matching the configuration used in MCQ. We will clarify this point in the revised version of our paper.
>
> **Q2: Comparison with VAVAE.**
>
> Thank you for the suggestion. We will include VAVAE for comparison in Table 2. However, we would like to clarify that VAVAE should not be considered a baseline for UniTok, as it is a continuous VAE tokenizer evaluated within the diffusion framework, whereas UniTok is a discrete VQVAE tokenizer evaluated under the autoregressive framework. This fundamental difference makes the two approaches less directly comparable. Instead, LlamaGen is a widely adopted baseline for discrete tokenizers. In Table 2, we strictly follow the LlamaGen implementation and only substitute the tokenizer with UniTok for evaluation. In this context, we believe that using LlamaGen as a baseline provides a more fair and meaningful comparison.
>
> **Q3: Why choose Liquid rather than VILA-U as the MLLM framework for evaluation?**
>
> - First, we would like to clarify that the earliest release of Liquid was on December 5, 2024, on arXiv, which is more than five months before the NeurIPS 2025 submission deadline.
> - Besides, the training data of VILA-U contains 15M internal data that are not publicly available. This makes it impossible to conduct controlled experiments with VILA-U.
> - We attempted to request access to the internal data used in both Liquid and VILA-U. However, only the authors of Liquid granted us access. Therefore, we selected Liquid as our evaluation codebase.
>
> **Q4: Does the performance in Table 3 come from better training data?**
>
> We agree that data quality has a significant impact on final performance. However, most unified models today use different data recipes for training and often involve internal data (e.g., as in Chameleon, VILA-U, Show-o, Janus, Liquid, among others). This makes direct, apple-to-apple comparisons impractical. Nonetheless, we believe the tokenizer innovation in UniTok is the primary contributor to the final performance, as demonstrated in Figure 3.
>
> **Q5: Evaluating UniTok on the GenEval benchmark.**
>
> Thank you for the suggestion. The table below presents a comparison of unified models on the GenEval benchmark. The overall trend is largely consistent with the results observed on GenAI-Bench, except that Show-o achieves better performance by scaling up the text-to-image training data to 2 billion samples. Notably, UniTok achieves non-trivial improvements over Liquid while using exactly the same set of text-to-image training data.
>
> | Method | # Data | Single Obj. | Two Obj. | Counting | Colors | Position | Color Attri. | Overall ↑ |
> |:---------|:-------:|:-------------:|:----------:|:----------:|:---------:|:---------:|:------------:|:----------:|
> | LWM | -- | 0.93 | 0.41 | 0.46 | 0.79 | 0.09 | 0.15 | 0.47 |
> | Chameleon | -- | -- | -- | -- | -- | -- | -- | 0.39 |
> | Show-o | 35M | 0.95 | 0.52 | 0.49 | 0.82 | 0.11 | 0.28 | 0.53 |
> | Show-o | 2.0B | 0.98 | 0.80 | 0.66 | 0.84 | 0.31 | 0.50 | 0.68 |
> | Janus | -- | 0.97 | 0.68 | 0.30 | 0.84 | 0.46 | 0.42 | 0.61 |
> | Liquid | 30M | 0.98 | 0.73 | 0.32 | 0.76 | 0.17 | 0.37 | 0.55 |
> | UniTok | 30M | 0.99 | 0.71 | 0.36 | 0.79 | 0.26 | 0.45 | 0.59 |
>
> #Data: Number of text-to-image data samples used in pretraining

---

> ### Author Response · Authors · 2025-08-06
> **Welcoming further discussions**
>
> Thank you for your feedback and for taking the time to review our response. We are glad that our reply addressed some of your concerns. If there are any remaining issues or questions, please let us know—we would be happy to provide further clarification.

---

### Official Review · Reviewer_uzhd · 2025-06-30

**Clarity:** 3
**Significance:** 3
**Originality:** 3
**Rating:** 5
**Confidence:** 3

**Summary:**

This manuscript focuses on developing effective tokenizer for unified understanding and generation. The authors analyze the potential problems of joint training with reconstruction and semantic supervision. Experiments show that the challenge lies in fabrication and discretization. To improve the vocabulary size and representation, the authors propose multi codebook quantization (MCQ) and utilize modified multi-head attention to replace the linear projection. The proposed method achieves excellent performance on both understanding and generation tasks.

**Questions:**

* The authors claim that MCQ can also boost the dimension of the vocabulary. However, the authors do not analyze or compare with the method that directly increases the dimension of a single codebook.

**Ethical Concerns:**

["NO or VERY MINOR ethics concerns only"]

**Final Justification:**

The rebuttal has well addressed my concerns. Thus, I would improve my rating to "Accept".

**Limitations:**

yes

**Quality:**

3

**Strengths And Weaknesses:**

- Strengths:
1. Tokenizer is a critical part for unified multi-modal models. This work propose an effective tokenizer for balancing understanding and generation performance. Comparison with most state-of-the-art methods demonstrate its effectiveness.

2. The motivation of addressing the poor representation of discrete tokenizer is clear. The proposed methods seem to reasonable.

3. The paper is well-written and easy to follow.

- Weaknesses:
1. The experiment results seem to inconsistent with the analysis. Specifically, the authors claim that the MQC is mainly designed to expand the vocabulary size. However, from Table 8 it can be seen that the performance increases as the number of codebook groups increases while the total vocabulary size remains unchanged. It is more like multi-head attention that utilizes more sub-space.

2. Figure 3 utilizes VQA score for analysis and draw the conclusion that the challenge of unified tokenizer is not the joint training. However, the authors claim in L144: "We observe a similar phenomenon where joint training results in sub-optimal ImageNet zero-shot classification accuracy and reconstruction FID compared to specialized training." It seems that the conclusions are inconsistent and are related to the tasks.

---

> ### Author Rebuttal · Authors · 2025-07-31
>
> Dear Reviewer uzhd,
>
> Thank you for your constructive feedback and thoughtful evaluation of our work. We hope our point-to-point response will address your concerns.
>
> **Q1: Clarification on the vocabulary size in Table 8**
>
> - First, we would like to clarify that the row of `Codebook` in Table 8 denotes the codebook size rather than the vocabulary size. (We apologize for any confusion caused by the notation.) In UniTok, the vocabulary size is exponentially larger than the codebook size. For example, UniTok uses 8 codebooks with 4096 code entries each, resulting in a total codebook size of 4096 × 8 = 32,768. In contrast, the vocabulary size is 4096^8 = 2^96, i.e., there are up to 4096^8 possible combinations of codes for each token.
>
> - We have extended Table 8 below to illustrate how both the vocabulary size and latent code dimension increase with the number of codebooks. As shown, increasing the number of codebooks simultaneously scales up the vocabulary size and latent dimension, which consistently leads to improved performance. We will revise Table 8 in our manuscript to make this relationship clearer.
>
> > | Codebook | 1×16384 | 2×8192 | 4×4096 | 8×2048 |
> > |:------------:|:----------:|:---------:|:---------:|:--------:|
> > | Vocabulary | 2^14 | 2^26 | 2^48 | 2^88 |
> > | Latent Dim. | 8 | 16 | 32 | 64 |
> > | rFID ↓ | 1.50 | 0.98 | 0.54 | 0.33 |
> > | Accuracy | 41.0% | 43.9% | 44.7% | 46.1% |
>
> **Q2: Inconsistency between observations in L144 and the conclusion**
>
> Although joint training can initially lead to a degradation in classification accuracy and reconstruction FID (L144), we observe that this degradation diminishes after improving the quantization methods (L147). For example, Table 6 shows that joint training actually results in better rFID and gFID. This supports our conclusion that VQVAE training and CLIP training do not inherently conflict; rather, the underlying issue is that discrete tokens lack sufficient capacity to simultaneously encode both low-level details and high-level semantics.
>
> **Q3: The impact of increasing the latent code dimension (single codebook setting)**
>
> - Thank you for your suggestion. The table below presents a comparison between the multi-codebook and single codebook settings, where the latent code dimension is scaled up to 64d. Due to time constraints, we trained the tokenizers on OpenImages using only the reconstruction loss. As shown, solely increasing the latent dimension of a single codebook results in lower codebook utilization rate (even with entropy loss) and degraded rFID.
>
> > | Codebook Size | Vocabulary Size | Latent Dim. | Codebook Utilization Rate | rFID ↓ |
> > |:-----------------:|:-------------------:|:-------------:|:------------------------------:|:-------:|
> > | 1×32768 | 2^14 | 64 | 82% | 2.20 |
> > | 8×4096 | 2^88 | 64 | 100% | 0.33 |
>
> - Besides, as mentioned in Lines 45-47, previous works have also reported similar effects when scaling the latent dimension. Specifically, Table 4 in ViT-VQGAN [1] provides a detailed and quantitative ablation study supporting this observation.
>
> [1] Yu, Jiahui, et al. "Vector-quantized image modeling with improved vqgan." arXiv preprint arXiv:2110.04627 (2021).

---

### Official Review · Reviewer_Qdyc · 2025-07-03

**Clarity:** 4
**Significance:** 3
**Originality:** 4
**Rating:** 6
**Confidence:** 4

**Summary:**

The paper introduces UniTok, a unified tokenizer designed to bridge the gap between visual generation and understanding. UniTok features a novel multi-codebook quantization mechanism that increases the tokenizer’s capacity and enables it to scale effectively. It achieves a record low reconstruction FID of 0.38 and 78.6% zero-shot accuracy on ImageNet. UniTok can be seamlessly integrated into multimodal large language models, enabling visual generation capabilities without compromising performance in understanding tasks.

**Questions:**

1) Can the tokenizer be enhanced by captioning loss (e.g., COCA)? This may enhance the downstream VQA task too.

**Ethical Concerns:**

["NO or VERY MINOR ethics concerns only"]

**Final Justification:**

I carefully read the authors' response, and my concerns are addressed. Thus I raise my score.

**Limitations:**

yes

**Quality:**

4

**Strengths And Weaknesses:**

**Strengths**

1) The paper is well-written and easy to read, with a logical flow that addresses both the technical details and experimental results effectively.
2) The paper presents a well-structured approach to visual tokenization, demonstrating solid experimental results. The multi-codebook quantization significantly improves performance over traditional VQ-based methods.
3) UniTok represents a important step forward in solving the problem of unified tokenization for both visual generation and understanding.
4) The use of multi-codebook quantization to effective.

Weakness:
1) The vision encoder is trained in 256x256 resolution. Will the LMM tasks in Table 6 need higher resolution? Can the author align the resolution when training the tokenizer?

---

> ### Author Rebuttal · Authors · 2025-07-31
>
> Dear Reviewer Qdyc,
>
> We greatly appreciate your insightful comments and support for our work. We hope that our responses below will address your questions.
>
> **Q1: Resolution used in LMM evaluation (Table 6)**
>
> In Table 6, we evaluate the model at 256×256 resolution for all tasks, consistent with the resolution used during UniTok training. We have also tested UniTok at 384×384 and 512×512 resolutions within the LLaVA framework. We found that UniTok generalizes well to 384×384 resolution without finetuning, likely due to its hybrid architecture (CNN + ViT). In particular, accuracy on the TextVQA benchmark—which benefits from higher resolution input—increases by 2%. However, further increasing the input resolution to 512×512 yields little additional improvement. We believe that finetuning UniTok at higher resolutions is necessary to make the most of high-resolution inputs.
>
> **Q2: Could UniTok be enhanced by incorporating captioning loss?**
>
> That's an excellent point. In addition to the image-text contrastive loss, image captioning loss is generally considered beneficial and is widely adopted in CLIP training, as demonstrated by works such as CoCa, BLIP2, and SigLIP2. Since UniTok inherits the CLIP training paradigm, we believe that incorporating captioning loss could further enhance UniTok's performance. Besides, TexTok [1] has shown that image captions can complement visual tokens to achieve better reconstruction quality and higher compression rates. Therefore, leveraging generated captions during the image decoding process could potentially yield additional improvements.
>
> [1] Zha, Kaiwen, et al. "Language-guided image tokenization for generation." Proceedings of the Computer Vision and Pattern Recognition Conference. 2025.

---

> > ### Comment · Reviewer_Qdyc · 2025-08-04
> >
> > Thank you for the thorough rebuttal. My thanks also go to the other reviewers, AC and PC for their efforts.
> >
> > Generalizing well to 384×384 resolution without finetuning is impressive!
> >
> > After reviewing your response and the other reviewers’ comments, I'll raise my score.

---

> > > ### Author Response · Authors · 2025-08-05
> > >
> > > Thank you very much for your thoughtful feedback and for taking the time to review our work! We sincerely appreciate your support and encouragement, as well as the constructive input from the other reviewers and AC.

---

### Official Review · Reviewer_rTfH · 2025-07-03

**Clarity:** 3
**Significance:** 3
**Originality:** 3
**Rating:** 4
**Confidence:** 3

**Summary:**

This paper proposes studying a visual tokenizer, **UniTok**, which can be effectively used for both generation and understanding tasks in MLLMs. The authors conduct extensive experimental analysis to investigate the importance of multi-codebook quantization and attention projection in their model design.

**Questions:**

Please see **Cons**

**Ethical Concerns:**

["NO or VERY MINOR ethics concerns only"]

**Final Justification:**

I have read the authors’ rebuttal. The paper presents a solid empirical study on using a visual tokenizer to unify visual understanding and generation tasks. I will maintain my initial score.

**Limitations:**

Yes

**Quality:**

3

**Strengths And Weaknesses:**

## Pros
* The writing is clear and easy to follow.
* The authors have conducted comprehensive experiments to analyze the bottlenecks of the tokenizer in unifying different tasks.
* The tokenizer demonstrates strong empirical performance in both generation and MLLM understanding tasks.


## Cons
These are not necessarily weaknesses, but addressing them may help further strengthen the paper:
* Table 6 shows that supervision from reconstruction loss and contrastive loss have different impacts on downstream tasks: adding contrastive loss further improves generation, but adding reconstruction loss often decreases performance on understanding tasks. Do the authors have any insights into why this occurs?
* Given the rapid development of this field, the authors might also consider including additional related work for discussion, such as [1].

[1] Hansen-Estruch, Philippe, et al. "Learnings from Scaling Visual Tokenizers for Reconstruction and Generation." *arXiv preprint* arXiv:2501.09755 (2025).

---

> ### Author Rebuttal · Authors · 2025-07-31
>
> Dear Reviewer rTfH,
> Thank you for your constructive feedback and thoughtful evaluation of our work. We hope that our responses below will address your questions and concerns.
>
> **Q1: How does contrastive loss improve generation?**
>
> In UniTok, the contrastive loss (CLIP supervision) injects semantic information into the code embeddings, which is conceptually similar to the MAE loss in MAETok [1] and the VF loss in VA-VAE [2]. Both of these works have demonstrated that **semantic guidance leads to more structured latent distributions**, which in turn facilitate diffusion-based generative modeling. Our experimental results in Tables 2 and 6 further confirm that this benefit extends to discrete latent spaces and autoregressive generative models.
>
> [1] Yao, Jingfeng, Bin Yang, and Xinggang Wang. "Reconstruction vs. generation: Taming optimization dilemma in latent diffusion models." Proceedings of the Computer Vision and Pattern Recognition Conference. 2025.
>
> [2] Chen, Hao, et al. "Masked autoencoders are effective tokenizers for diffusion models." Forty-second International Conference on Machine Learning. 2025.
>
> **Q2: How does reconstruction loss impact understanding?**
>
> Intuitively, reconstruction loss encourages the tokenizer to preserve low-level details (e.g., texture) when encoding images. Since the total representation capacity of discrete tokens is limited, this creates a trade-off — allocating more "storage space" to low-level details leaves less capacity for encoding high-level semantics, which can lead to degraded understanding performance. However, when the discrete latent space is enlarged (i.e., by using multi-codebook quantization), this effect becomes minor and has only a mild impact on understanding performance.
>
> **Q3: Related work ViTok**
>
> Thank you for the suggestion. ViTok presents a comprehensive study on the scaling properties of VAEs, offering valuable insights into VAE design. We appreciate your recommendation and will include a discussion of this work in our revised manuscript.

---

### Decision · Program_Chairs · 2025-09-17

**Decision:**

Accept (spotlight)

**Comment:**

UniTok studies a unified visual tokenizer that works for both generation and understanding by diagnosing that the apparent conflict between reconstruction and semantic supervision stems from the limited capacity of discrete token spaces rather than the losses themselves. The paper proposes multi codebook quantization with an attention based projection to expand both the effective vocabulary and the latent bottleneck, and shows state of the art image generation and recognition outcomes, including a reconstruction FID of 0.38 and 78.6 percent zero shot ImageNet accuracy, while enabling native generation inside MLLMs without harming understanding. It further reports large gains in CFG free generation, cutting gFID on ImageNet 256 by an order of magnitude, and backs the claims with broad ablations and analyses. Strengths are a clear problem formulation, a principled design that links analysis to method, extensive experiments across both modalities, and compelling practical integration into an MLLM. Weaknesses raised in review include fairness and coverage of baselines such as RQ settings and VAVAE, resolution alignment for LMM evaluation, reliance on Liquid rather than VILA U for some comparisons, possible data confounds in one table, and a request to include GenEval; there were also questions about why contrastive supervision aids generation and how reconstruction affects understanding capacity. Overall the AC find the technical contribution substantial and clean, the empirical evidence strong, and the paper well written and easy to follow.

During rebuttal the authors answered the main points thoroughly. For Reviewer rTfH, they explained that contrastive supervision injects semantics into code embeddings and that reconstruction can trade off high level capacity when the discrete bottleneck is too tight, which their larger multi codebook space alleviates; Reviewer rTfH kept a borderline accept after acknowledging the responses. For Reviewer Qdyc, they clarified that all LMM results used 256 by 256 to match training and reported successful generalization to 384 by 384 with a two point gain on TextVQA and little benefit at 512 by 512; Reviewer Qdyc raised to strong accept. For Reviewer uzhd, they disambiguated that Table 8 varied codebook groups while exponentially growing the combinatorial vocabulary and latent dimension, added an ablation showing that simply widening a single codebook to 64d reduces utilization and hurts rFID, and reconciled the joint training discussion by showing improved outcomes once quantization is fixed; Reviewer uzhd raised to accept. For Reviewer emrz, they clarified the multi code setting used for RQ, argued why VAVAE is not a directly comparable discrete baseline while agreeing to add it, justified choosing Liquid due to data access constraints, and provided a GenEval table where UniTok improves over Liquid under the same training data. Weighing these, the most important reasons for acceptance are that the paper isolates a widely observed failure mode to a concrete capacity bottleneck, proposes a simple and general fix that materially advances both generation and understanding in one tokenizer, and validates it across strong baselines with careful ablations and practical MLLM integration. Hence, the AC recommends acceptance.